# X6: A Novel Antibody for Potential Use in Gluten Quantification

**DOI:** 10.3390/molecules25143107

**Published:** 2020-07-08

**Authors:** Aleksandrina Shatalova, Ivan Shatalov, Yuri Lebedin

**Affiliations:** 1Faculty of Food Biotechnologies and Engineering, ITMO University, Lomonosova Street, 9, 191002 St. Petersburg, Russia; shatalovaaleks@mail.ru; 2ELISA Development Department, XEMA Co., Ltd., Degtyarniy Lane, 8-10 St., 191144 Petersburg, Russian; 3Head Office, XEMA Co., Ltd., 9-th Parkovaya str., 48, 105264 Moscow, Russian; lebedin@xema-medica.com

**Keywords:** gliadin, epitope mapping, ELISA, celiac disease

## Abstract

Gliadin is a fraction of gluten, known to trigger celiac disease in susceptible people. To date, the life-long gluten-free diet is used for the prevention of this disease. Hence, methods for gluten control in foods are of significant importance. Being one of the most-used methods used for this purpose, ELISA should use high-affinity antibodies to gliadin peptides involved into celiac process. This study investigates the characteristics of a novel anti-gliadin antibody X6. We found the QXQPFPXP site to be a recognized epitope that provides specific binding of the antibody to cereal prolamins involved in celiac disease manifestation. A specificity study using immunoblotting shows the recognition of wheat, barley and rye proteins—as well as α-gliadin homologs from non-edible cereals (*Dasypyrum villosum*). Reactivity to avenin was less pronounced, as this protein does not contain the PFP motif most critical for antibody recognition. The proteins of *Zea mays* and *Setaria italica* were not recognized by X6. X6-based ELISA highly correlated with R5 and G12, which are Codex Alimentarius standards in the quantitative assessment of gluten content (Pearson’s R = 0.86 and 0.87, respectively). Qualitative assessment revealed no significant differences between R5 and G12 and X6.

## 1. Introduction

Celiac disease is a chronic inflammatory disorder of the small intestine and is mediated by gluten intake immune response in susceptible people [1]. Gluten is a wheat grain storage protein; it consists of two main fractions: glutenin—a mixture of proteins soluble in dilute acids or alkalis and gliadin—alcohol-soluble fraction [2,3,4]. Gliadins play a major role in the celiac disease manifestation. Scientists found the homologs of gliadin proteins in other cereals: in oats (avenin), barley (hordein), rye (secalin), and they are able to trigger the celiac disease process [5,6,7].

Most cases of celiac disease occur in patients with the HLA-DQ2/-DQ8 haplotype, which is the main risk-factor [8,9]. Researchers discovered the gliadin fragments deamidated by tissue transglutaminase (tTG) increase affinity for DQ2/-DQ8 heterodimeric surface receptors. Therefore, the reaction of gliadin fragments with tTG is also one of the key factors in the formation of the pathologic process [10].

In the 2000s, scientists described some of the immunodominant gliadin peptides associated with the pathogenesis of celiac disease. In particular, the α9-gliadin (57–68) QLQPFPQPQLPY and α2-gliadin (62–75) PQPQLPYPQPQLPY belong to them [11]. It was revealed that fragments rich in proline and glutamine are highly resistant to the action of digestive enzymes [12]. Scientists detected a proteolytically stable fragment of α2-gliadin, which is 33-mer LQLQPFPQPQLPYPQPQLPYPQPQLPYPQPQPF peptide (fragment 57–89) while conducting in vitro experiment on this protein digestion. The fragment includes both peptides mentioned above. This 33-mer peptide is more prone to deamination by tTG and, accordingly, is the most important trigger of the inflammatory process in patients with celiac disease [13].

Currently, the most effective way to prevent celiac disease is a gluten-free diet. The complete exclusion of gluten and homologous proteins from the diet of patients leads to persistent remission [14]. Therefore, a very important area in food analysis is the development and evolution of methods for the qualitative and quantitative detection of gluten and immunogenic peptides in food products.

There are many methods for qualitative and quantitative evaluation of gluten content detection based on chromatography and mass spectrometry [15,16], such as MALDI-TOF and polymerase chain reaction [4,17]. Despite such advantages as high sensitivity and specificity of recognition, these methods are much less spread than the enzyme-linked immunosorbent assay (ELISA) [4,7].

There are various ELISA test systems designed to detect and quantify gluten in food. Table 1 illustrates the specificity of the most characterized and commercially available antibodies to gliadin.

Skerritt et al. obtained the monoclonal antibodies—401.21, which were initially used to detect gluten in food [19]. Such antibodies were mainly specific for HMW and LMW glutelin, ω-gliadin, and had a mild cross-reaction with α- and γ-gliadin. These antibodies were less reactive with barley prolamins, but more reactive with rye prolamins.

At the same time, a test system based on these antibodies was applied as AOAC Official Method 991.19 for the detection and quantification of gluten in food products in 1995 [20]. Later, in 2006, the Codex Alimentarius Commission approved a test system based on R5 antibody as a method for determining gluten in gluten-free products [21]. These antibodies were able to perfectly recognize α- and γ-gliadins, but also had the increased reactivity with barley prolamins. Nevertheless, there was a cross-reactivity with proteins of lupine and soy at the initial stages, which was eliminated by the improvement of methods for the prolamins extraction from a food matrix [22]. It is worth noting that the AOAC Research Institute accepted the test systems based on R5 antibody as an official method in 2012. Along with the R5 ELISA, the official method for controlling gluten in food, there is also a method based on the use of G12-based ELISA [23]. This antibody has specificity to the toxic 33-mer. A distinctive feature of G12 antibody is the ability to recognize those varieties of oats with prolamins acting as a trigger for celiac disease [7].

The analyte extraction method has also an impact on the quality of the gluten content analytical evaluation. During the food processing, proteins, particularly gluten, undergo a number of changes forming strong molecular scaffolds [24]. Therefore, the extraction and subsequent detection of these proteins require operations to increase their solubility. As noted above, prolamins have excellent solubility in alcohol solutions, but their alcohol extraction from food matrices is difficult. Accordingly, in 2005 Mendez et al. proposed an extraction method based on the use of a buffer solution that consists of 2-mercaptoethanol, which reduces the disulfide bonds of a dough matrix and chaotrope (guanidine), which increases the solubility of the prolamin fraction in aqueous systems. The resulting buffer—cocktail solution—provides an almost twofold increase in the extractability of prolamins compared to conventional alcohol extraction [25]. Due to this, Codex Alimentarius Commission uses the Mendez–R5 method as the official method for determining the gluten in food [21].

Therefore, to meet the requirements for gluten-free foods control, an ELISA to apply has to possess not only high analytical characteristics (limit of quantification (LOQ) and limit of detection (LOD)) to recognize precisely celiac-related epitopes, but also the antibodies and components of test systems to be compatible with modern methods of gluten extraction.

## 2. Materials and Methods

Reagents: Gliadin from wheat (G3375, Sigma-Aldrich, Moscow, Russia), α2–gliadin 33-mer fused with carrier protein (G052, ZEDIRA GmbH, Darmstadt, Germany), γ-gliadin 26–mer fused with carrier protein (G051, ZEDIRA GmbH, Darmstadt, Germany), guanidine hydrochloride (Gu-HCl, Scharlau, Scharlab, Barcelona, Spain), 2-mercaptoethanol (2ME, Amresco, Solon, OH, USA). Anti-wheat gliadin monoclonal antibody, 28 clones (XGY1-3, X4, XGY5, X6, XGY7-9, X10, XGY11–26, XH1 and XH4, mAbs), rabbit anti-mouse IgG (As302, conjugate), wash buffer (S008), antibody coating buffer (S000), blocking buffer (S002), substrate solution (R055), ELISA buffer (S012), gliadin sample buffer (S380), 96-well microplates high sorption (N001) were from XEMA Co., Ltd. (Moscow, Russia).

Food Samples: Different types of food samples with unknown concentration of gluten were collected from local supermarkets. Apparently gluten-positive samples were not under consideration. The samples were divided into 5 groups: unprocessed (*n* = 18), snack (*n* = 18), bakery (*n* = 16), flour ((*n* = 16) and high-gluten samples (*n* = 12). The samples were extracted and analyzed according to the standard procedure described below. All the samples were analyzed in duplicates. The detailed list of the samples used, and data obtained is given in “Appendix A”.

### 2.1. Antibody Production

All antibodies used in our experiments were developed by XEMA, Co. Ltd (Moscow, Russia).

All cell lines producing antibodies used in our experiments were obtained from the spleens of immunized 8 week female BALB/c mice fused with Sp2/0 murine myeloma according to standard hybridoma technique described elsewhere [26]. To obtain the X/XGY antibodies, wheat gliadin was used as an immunogen; XH antibodies were raised against barley hordein. gliadin and hordein for immunization were obtained by the company using ethanolic extraction from local varieties of *Triticum aestivum* and *Hordeum vulgare*, respectively.

All antibodies were cultured in ascitic fluids in 6 week female BALB/c mice and purified using affinity chromatography on protein G-sepharose (IgG-subclasses) (GE Healthcare, Uppsala, Sweden) or protein L affinity resin (IgM subclass) (Tosoh Bioscience, Tokyo, Japan) according to Manufacturers’ instructions.

### 2.2. Antibody Selection

A selection of 96-well plates were coated overnight with 100 µL/well of fused 26-mer or 33-mer dissolved at 5 µg/mL in the coating buffer, pH 9.6. The plates were subsequently blocked with casein-base coating solution at 200 µL/well for 3 h at RT and then dried overnight at RT. As a negative control, a carrier protein (G055, ZEDIRA GmbH, Darmstadt, Germany) was used. It was sorbed in microplates according to the same procedure. mAbs were dissolved in ELISA buffer at 5 µg/mL and added to wells at 100 µL/well in duplicates. The reaction proceeded at 37 °C for 30 min and then washed 3 times with the wash buffer. Then, the conjugate solution in ELISA buffer was added into wells at 100 µL/well and the plate were left at 37 °C for 30 min. After 5-time washing the substrate solution was added to the wells at 100 µL/well. Color development was allowed to proceed for 15 min at RT. The reaction was stopped by the addition of 0.5-M sulfuric acid and the optical density in the wells was measured at 450 nm spectrophotometrically (Epoch, BioTek, Winooski, VT, USA). For each antibody, the signal-to-noise ratio was calculated according to the Formula (1):(1)$$S/N\;\mathrm{ratio}=\frac{{\bar{\mathrm{OD}}}_{\mathrm{peptide}}}{{\bar{\mathrm{OD}}}_{\;\mathrm{carrier}}}$$
where $\bar{\mathrm{OD}}$ is the mean OD value of duplicates for the reaction with the corresponding sample. Within the assay ${\bar{\mathrm{OD}}}_{\;\mathrm{carrier}}$ value did not exceed 0.12 U. Antibodies giving the S/N ratio of more than 5 with both peptides were considered for further experiments.

### 2.3. Epitope Mapping

Epitope mapping was performed at PEPperPRINT GmbH, Germany. Briefly, the epitope-containing peptide was subjected to subsequent amino-acid replacement, and the resulting peptides (240 peptides in triplicates) were printed onto a chip (Microarray). After the blocking step, the resulting microarray was incubated with the antibody at the concentration of 1 µg/mL followed by the staining with secondary goat anti-mouse IgG (H + L) DyLight680 antibody. The read-out was performed with a LI-COR Odyssey imaging system at scanning intensities of 7/7 (red/green). Quantification of spot intensities and peptide annotation were based on the 16-bit grayscale tiff files at scanning intensities of 7/7 (red/green) that exhibit a higher dynamic range than the 24-bit colorized tiff files. Microarray image analysis was done with PepSlide^®^ analyzer (SICASYS Software GmbH, Heidelberg, Germany). The data obtained is listed in Appendix A.

### 2.4. ELISA Pre-Construction

All antibodies were labeled with horseradish peroxidase (HRP) using the periodate-oxidation procedure described elsewhere. gliadin was serially dissolved in 60% ethanol and the gluten content of the solutions was measured with R5-based ELISA (Ridascreen, R-Biopharm AG, Darmstadt, Germany). These solutions were used for calibration of the X6-based ELISA.

96-well plates were coated with antibodies as described earlier. Sandwich pairs formation was screened with the gliadin, dissolved in gliadin sample buffer (S380, XEMA, Co. Ltd, Moscow, Russia) at 20-ng/mL (positive sample) and incubated at 37 °C for 30 min. For background evaluation, gliadin sample buffer was used (0-ng/mL gliadin, negative sample). The positive and negative samples were applied into the wells in duplicates.

HRP-labeled antibodies dissolved in ELISA buffer 1:50,000 were used as tracer antibodies. The incubation was conducted at 37 °C for 30 min. After washing 5 times, the substrate solution was added to the wells at 100 µL/well. Color development was allowed to proceed for 15 min at RT. The reaction was stopped by the addition of 0.5-M sulfuric acid and the optical density in the wells was measured at 450 nm spectrophotometrically (Epoch, BioTek, Winooski, VT, USA). The S/N ratios were calculated using the Formula (2):(2)$$S/N\;\mathrm{ratio}=\frac{{\bar{\mathrm{OD}}}_{20\;\mathrm{ng}/\mathrm{mL}}}{{\bar{\mathrm{OD}}}_{\;0\;\mathrm{ng}/\mathrm{mL}}}$$
where $\bar{\mathrm{OD}}$ is the mean OD value of duplicates of the positive and negative samples, respectively. Antibody pairs having the highest signal-to-noise ratio (S/N ratio) were used for ELISA construction after adjusting the buffer composition, conjugate concentration and procedure conditions. The final characteristics of the method are described below.

### 2.5. Determination of ELISA Characteristics

The calibration curve was built with 6 points having 0, 5, 10, 20, 40, 80 ng/mL of gliadin using 4-PL approximation model. The following characteristics were evaluated for the test system: limit of detection, limit of quantification, linearity and recovery.

Limit of detection (LOD) was assessed according to the procedure described earlier using the Formula (3) [27]:(3)$$LOD={\bar{\mathrm{OD}}}_{0\;\mathrm{ng}/\mathrm{mL}}+1.645\times {\mathrm{SD}}_{0\;\mathrm{ng}/\mathrm{mL}}+1.645\times {\mathrm{SD}}_{5\;\mathrm{ng}/\mathrm{mL}}$$
where $\bar{{\mathrm{OD}}_{0\;\mathrm{ng}/\mathrm{mL}}}$ and $\bar{{\mathrm{SD}}_{0\;\mathrm{ng}/\mathrm{mL}}}$ are the mean OD and standard deviation (SD) value of 20 repeats of blank calibrator (0-ng/mL gliadin) respectively; $\bar{{\mathrm{SD}}_{5\;\mathrm{ng}/\mathrm{mL}}}$ is SD value of 5 repeats of lowest calibrator.

Limit of quantification (LOQ) was determined according to the Equation (4) [28,29]:(4)$$LOQ={\bar{\mathrm{OD}}}_{0\;\mathrm{ng}/\mathrm{mL}}+10\;{\mathrm{SD}}_{0\;\mathrm{ng}/\mathrm{mL}}$$
where $\bar{{\mathrm{OD}}_{0\;\mathrm{ng}/\mathrm{mL}}}$ and $\bar{{\mathrm{SD}}_{0\;\mathrm{ng}/\mathrm{mL}}}$ are the mean OD and SD value of 20 repeats of blank calibrator (0-ng/mL gliadin) respectively.

For the linearity test, a calibration sample with a maximum concentration (80 ng/mL) was diluted serially at 2-, 4-, 8- and 16-times with gliadin sample buffer and the resulting concentration was assayed. The linearity of the test system was calculated using the Equation (5):(5)$$\mathrm{LINEARITY}=\frac{C_{\mathrm{PREDICTED}}-C_{\mathrm{OBTAINED}}}{C_{\mathrm{OBTAINED}}}\times 100\%$$
where C_PREDICTED_ and C_OBTAINED_ are the predicted and obtained gliadin concentration in sample, respectively.

For the recovery test, a calibration sample 0 ng/mL + 20 ng/mL, 10 ng/mL + 40 ng/mL, 80 ng/mL + 10 ng/mL were mixed (1:1), giving the predicted concentrations of 10, 25 and 45 ng/mL, respectively. The sample were assayed, and the data obtained were applied to the Equation (6):(6)$$\mathrm{RECOVERY}=\frac{C_{\mathrm{PREDICTED}}-C_{\mathrm{OBTAINED}}}{C_{\mathrm{OBTAINED}}}\times 100\%,$$
where C_PREDICTED_ and C_OBTAINED_ are the predicted and obtained gliadin concentration in sample, respectively.

### 2.6. Gluten Quantification in Prolamins

Prolamins were extracted with ethanol from corresponding cereals. Grinded samples of *T. aestivum*, *A. sativa*, *S. cereal*, *H. vulgare* were extracted with 60% ethanol (1:10). protein concentration was determined according to Pierce BCA method (Thermo Scientific, Rockford, IL, USA) with BSA as standard. Before protein concentration measurement a part of each sample was prediluted with PBS 1:10 to avoid the interference with the analysis and immediately subjected to the procedure. Afterwards, the concentration of each extract was brought to 1 mg/mL with 60% ethanol and subjected to serial dilutions in gliadin sample buffer followed by ELISA.

### 2.7. Evaluation of Antibody Specificity via Electrophoresis and Western-Blotting

To assess the composition of extracts we used SDS–PAGE in reducing (with 2ME) conditions according to Lemmli method in a mini–chamber (BioRad Laboratories Inc, Moscow, Russia).

Grinded cereals (*Triticum urartu*, *Triticum turgidum*, *Triticum aestivum*, *Secale cereale*, *Hordeum vulgare*, *Avena sativa*, *Zea mays*, *Dasypyrum villosum*, *Setaria italica*) were extracted with 60% ethanol with the addition of 0.7-M 2ME. The extraction was completed in 3 h at RT under continuous shaking. The extracts were mixed with Sample buffer at 1:3 and applied onto the gel at 20 µL/well.

#### 2.7.1. Electrophoresis

The 12% polyacrylamide gel was used. The buffer solution used for electrophoresis was Tris-glycine, pH 8.3, containing 0.1% sodium dodecyl sulfate (SDS). The 0.06-M Tris-HCl buffer, containing 0.1% SDS, glycerol and bromophenol blue, pH 6.8 was used as a sample buffer.

Electrophoresis was performed for 2 h at a current of 30 mA. The starting voltage was 40 mV, which was increased to 150 mV after the sample entered the upper gel. After the front of the samples reached the middle of the gel the voltage was increased to 210 mV. After completing the run, the gels were fixed with the fixing solution composed of 40% methanol and 10% acetic acid for 30 min, then they were washed three times with deionized water to remove excessive SDS. The gel was stained with 0.1% Coomassie brilliant blue R250 in 40% methanol and 10% acetic acid. To decrease background the stained gels were washed with 10% acetic acid overnight with constant agitation at 400 rpm (ST-3L, ELMI, Riga, Latvia).

#### 2.7.2. Western-Blotting

After electrophoresis, the gels were soaked into transfer buffer (TransBlot, BioRad Laboratories, Moscow, Russia) and then transferred onto PVDF-membrane using semi-dry transferring system (TransBlot Turbo, BioRad Laboratories, Moscow, Russia ) according to the manufacturer’s protocol. The transfer was conducted at 1.3 A and 25 V for 10 min. Afterwards, the membrane was blocked with the blocking buffer for 30 min at 37 °C. After the blocking step, X6 antibody-HRP conjugate dissolved in the gliadin sample buffer was incubated with the membrane for 30 min at 37 °C followed by 5-time washing with the wash buffer (each step 5 min). The washed membrane was incubated in the substrate solution, containing 0.05% of sodium nitroprusside. The development of bands was proceeded for 15 min. The membrane was washed once with the wash buffer to reduce background.

### 2.8. MALDI-TOF

Protein bands that were reactive in western blot were cut from the gel for MALDI-TOF analysis. The procedure was conducted at the Institute of Biomedical Chemistry, Moscow. All reagents were purchased at Sigma-Aldrich (Moscow, Russia).

The procedure was described previously [30]. Briefly, the gel cuts were washed twice with 50% acetonitrile solution in 0.1-M NH_4_HCO_3_ for 20 min at 37 °C followed by dehydration with acetonitrile for 5 min. The tryptic digestion was performed with 5 μL of enzyme in 0.1-M NH_4_HCO_3_ for 4 h at 37 °C with subsequent peptide extraction with 0.7% TFA. The extracts obtained were analyzed using MALDI-TOF mass spectrometry.

To obtain mass spectra of digests MALDI-TOF/TOF mass-spectrometer (Ultrafle II, Bruker Daltonics, Germany) equipped with an Nd:YAG laser in the reflector mode was used. The measurement of monoisotopic [MH+] ions was conducted in the 700–4500 *m/z* range with a tolerance of 70 ppm. The lift mode was used to obtain fragment ion spectra. The accuracy of fragment ion mass peak measurements was within 1 Da.

Spectral data analysis was performed via FlexAnalysis 3.3 software (Bruker Daltonics, Germany). To identify individual proteins MASCOT search software. Reliably identified proteins had the scores of >82 (*p* < 0.05) with the use of “peptide fingerprint” option and >55 (*p* < 0.05) with the use of “ion score” option. The search was conducted in National Center for Biotechnology Information (NCBI) databases or EST (expressed sequence tag) plant database or both.

### 2.9. Gluten Measurement in Foods with ELISA

For comparison experiments we used both R5-based (Ridascreen, R-Biopharm AG, Darmstadt, Germany) and G12 (AgraQuant, Romer Labs, Runcorn, UK) ELISA kits. Commercial extraction procedures were replaced with an in-house modified extraction method based on cocktail and ethanol combination, which is described below. The extracts were assayed simultaneously in parallel with three test kits: G12-, R5- and X6-based according to the procedure given in Table 2.

Food samples were milled in a food grinder (J500, Bork, Moscow, Russia). A total of 0.25 g of homogenized material was mixed with 2.5 mL of patented Mendez cocktail solution (EP 2003448 A1) (2-M GuHCl + 0.25-M 2ME in PBS, pH 7.3). The mixtures were incubated in a water bath (Biosan) at 50 °C for 40 min. After the incubation was completed 7.5 mL of 80% ethanol was added to the samples and prolamin fraction was solubilized for 1 h at room temperature. The samples were centrifuged at 4000× *g* (Eppendorf, Hamburg, Germany) for 10 min. For further analysis, the extracts obtained were used according to the procedure, described in Table 2.

### 2.10. Statistical Analysis

Gliadin concentrations were determined with 4PL interpolation method. Statistical analyses were performed in R programming environment.

For correlation analysis the data were log-transformed, and normality was evaluated using the Shapiro–Wilk test of the shapiro.test package. The correlation analysis was conducted using the package cor.plot. The ROC-analysis was performed using a plotROC package. The McNemar test from the mcnemar.test package was used to identify qualitative differences. Data visualization was done using ggplot package. A Mann–Whitney test was performed using ggpubr::compare_means package.

## 3. Results and Discussion

### 3.1. PFP Motif Is Critical for Antibody Recognition

When choosing antibodies for the experiment, we used the data of the XEMA company on the binding of anti-gliadin antibodies to proteins of other grains. To determine the ability to bind to 26-mer and 33-mer gliadin, we selected only antibodies of class IgG, which recognize barley and rye proteins and have no cross-reactivity or have it negligible with corn proteins. After performing the reaction with the peptides, three antibodies, X4, 6, 10, were selected because they can recognize both immunogenic peptides and meet the criteria described above. The list of antibodies characteristics used for the experiment is provided in Appendix A.

We used selected mAbs to construct the ELISA test system of the “sandwich” type. The data (ODs and S/N ratios) are presented in Table 3.

As a result, we found that the best signal-to-noise ratio was obtained for the X6/X6-HRP pair and conducted further studies only with this antibody. Based on the amino acid structure, we found that both peptides contain a repeating fragment—QPFPQ. Therefore, we can suggest that it is part of the epitope. For the subsequent peptide mapping, we selected a shorter peptide containing the indicated fragment—QLQPFPQPQLPY. Introducing amino acid substitutions in each of the 12 positions resulted in the chip containing the formed peptides (*n* = 720). The reactivity of X6 antibody with the obtained set of peptides is presented in Figure 1.

We established that substituting at specific positions led to both a decrease in the antibody’s recognition of the obtained peptides (for example, positions 4–6) and an increase in affinity for this one (position 2). In the latter case, there was a strict dependence on a specific amino acid, which was replaced. As a result of heat map visual analysis, we assumed that the epitope for the antibody studied was between positions 1–8, while positions 9–12 do not affect the antibody recognition.

Figure 2 shows the comparison results of deviations in binding to peptides when a substitution is made at a specific position. Amino acid substitutions at positions 4–6 lead to the most significant decrease in the intensity of antibody binding to peptides. In this case, there is an almost complete loss of the antibody affinity for the resulting peptides. The substitutions at positions 1, 3 and 8 also mainly resulted in an almost 4-fold decrease in the intensity of antibody binding to these peptides. In this case, position 1 was more sensitive, since substitutions in it caused a 75% decrease in affinity (16 cases out of 19), whereas antibody affinity with substitutions at positions 3 and 8 decreased also in 13 and 14 cases out of 19, respectively.

Based on the data presented, we have established that the desired epitope is a QXQPFPXP peptide and the PFP part is critical for antibody recognition.

### 3.2. X6 Is Specific to Proteins Involved into Celiac Manifestation

During the protein specificity studying via immunoblotting (Figure 3b), we found that this antibody was able to bind to the prolamins of *Triticum urartu*, *T. turgidum*, *T. aestivum*, *Secale cereale*, *Hordeum vulgare*, *Avena sativa* and *Dasypyrum villosum*, but did not recognize *Zea mays* or *Setaria italica* prolamins. The first 5 representatives of cereal crops are involved in the manifestation of celiac disease.

Regarding *D. villosum*, according to GenBank, the α-gliadin sequence of it (AIY27554.1) contains a repeating QPQPFP fragment, which is obviously, bound with.

While studying of the specificity by immunoblotting, we found that all cereal proteins with QXQPFPXP epitope sequence can be recognized by the X6 antibody, and the simultaneous presence of Q1 and P8 is not necessary for recognition. On the other hand, the avenin sequence contains no PFP part, which absence is critical for antibody recognition, as noted above. This leads to a low antibody affinity to this protein. Table 5 gives the measured ppm concentrations for various prolamins preparations at a concentration of 1 mg/mL.

The gluten concentration in the extract from oats is more than 900 times lower than the concentration of gluten determined in extracts of wheat, rye and barley, according to the results obtained by the X6 antibody-based test system. We can explain this observation by the lower antibody specificity to avenin compared to other prolamins.

### 3.3. X6-Based ELISA Is Similar to Codex Alimentarius Standards

Enzyme-linked immunosorbent assay (ELISA) is the most common screening tool to determine the presence of gluten in foods. The method is characterized by high sensitivity and specificity facilitated by the high affinity of antigen–antibody interaction and features of the specific analytical method applied. Currently, the products containing no more than 20 ppm gluten in their composition are considered “gluten-free”. Therefore, to avoid false results, test systems used for food analysis should be carefully calibrated in the range of up to 20 ppm.

We have developed a simple “sandwich” type ELISA test system. Calibration samples were prepared using Sigma gliadin and were nominated relative to the Codex Alimentarius standard test system based on R5 antibodies. By performing 20 measurements, we found that LOD and LOQ are 2.5 and 5 ppm, respectively. At the same time, linearity and recovery of gliadin concentration were in the range of 90–110%. Due to the great heterogeneity of its composition, antibodies are able to recognize various gluten-containing samples differently depending on the method used. To assess the correctness of determining the gluten content in food products, we performed cross-validation between the X6, R5 and G12 antibodies-based system. According to the type of products, there were three groups present in the set: bakery (*n* = 16), snacks (*n* = 18), flour (*n* = 16). We also considered products with a high gluten content (*n* = 12) to evaluate the test system behavior at a high analyte concentration, as well as the quality of the ELISA response to analyte dilution. The results are presented in Figure 4.

In the “bakery” and “flour” groups, most of the samples were gluten-free products (less than 20 ppm). There were no significant differences in these groups measuring the gluten content by the three methods. When comparing the gluten content in the product groups, significant differences were observed only in the “snack” group between the test systems based on G12 and X6. This group included mainly gluten-free products, so the features of the test calibration in this range (up to 20 ppm) and the result of weakly expressed nonspecific binding to specific antibodies may cause the obtained differences.

To establish how the differences influence on the interpretation of the gluten measurement result in the sample and the product classification as the corresponding type according to the gluten content, we conducted the McNemar’s test for this group of products. Table 6 shows the measured gluten concentrations in the “snack” group.

As a result of the test, it was found that the differences obtained are not significant. All the ELISA identify products equally.

To assess a correlation between the data obtained using the X6-based test system and the data obtained by existing methods with antibodies G12 and R5, we performed a correlation analysis. All the three test systems have different analytical characteristics, namely, the dynamic range and the limit of determination. We excluded products with values obtained less than five parts per million of gluten and containing more than 80 ppm from the correlation analysis (see Appendix A). The gluten content in the remaining samples (19 in total) was log-transformed for normalization. The results of the analysis are presented in Figure 5. The concentration values obtained for the X6-based test system highly correlated with the data obtained by the reference test systems using R5 and G12 antibodies (Pearson’s R = 0.87 and 0.86, respectively). This generally indicates a high similarity of the presented determination method with the reference one.

In conclusion, we estimated the X6-antibody-based test system analytical characteristics for using it as a tool for screening gluten-containing products. The evaluation was performed by constructing ROC curves and the areas under the curves (AUC) were assessed. In this case, the data obtained with R5-based ELISA were used as reference. Figure 5b shows the resulting ROC curves. From the analysis of ROC curves, it follows that the gluten content qualitative results in the samples obtained using various test systems highly correlate. In particular, the area under the curve, constructed according to the data of X6-based ELISA, completely coincides with G12-based one and tends to 1. This indicates the practical interchangeability of all three methods when screening gluten-free products. Table 7 demonstrates the results of the McNemar’s test for all the products examined that we performed.

The McNemar’s test revealed no significant differences in determining the gliadin content of the test system based on X6 and existing reference ones (*p* > 0.05). It is worth noting that during the selection analysis there were almost complete coincidence of the gluten determination results in samples when comparing test systems based on X6 and G12; in this case, only one sample out of 80 is defined as negative at the reference. It was also found that none of the negative samples on R5 was determined as positive on X6, which indicates the high specificity of this antibody.

## 4. Conclusions

In this study, we have characterized a new anti-gliadin antibody X6 and have estimated its potential for gluten content evaluation test systems. The antibody is specific for cereal proteins, with its affinity highly dependent on the presence of a PFP part in the protein, which is the most critical for antibody recognition. The structurally recognizable epitope is similar to R5; but larger in size, which can potentially lead to a higher specificity. According to the comparison results of the reference test systems and the X6 antibody-based ELISA we found their almost complete coincidence.

Our data allow us to conclude that the X6 antibody has a potential to be used in sandwich-type ELISA test systems designed to determine the gluten content in foods.

## Figures and Tables

**Figure 1 molecules-25-03107-f001:**
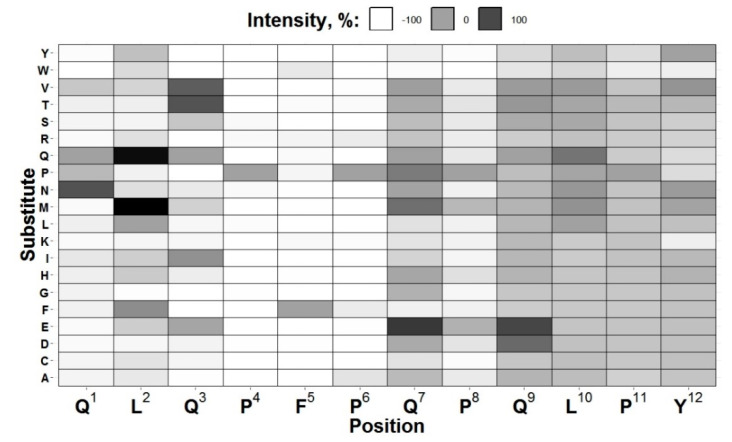
X6 antibody reactivity with the studied peptide set heat map. The color intensity is proportional to the intensity of the antibody binding to a particular peptide. PFP region is almost white resulted from its complete intolerance to amino acid replacement.

**Figure 2 molecules-25-03107-f002:**
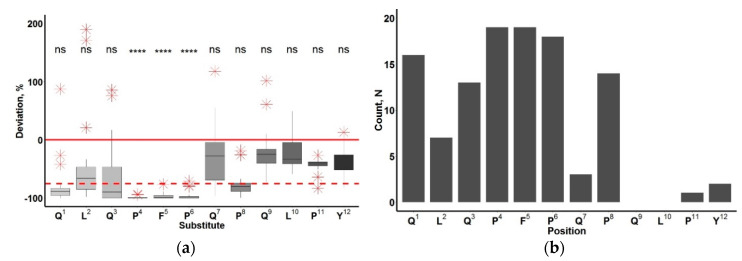
Quantitative change in the binding of an antibody to the QLQPFPQPQLPY peptide upon introduction of amino acid substitutions. (**a**) percent deviation of affinity from the original peptide; outliers are marked with a red asterisk, Mann–Whitney test significance levels are black: ns: *p* > 0.05; ****: *p* ≤ 0.0001; (**b**) number of amino acid substitution variants in the peptide, leading to a 75% decrease in the affinity of the antibody to it.

**Figure 3 molecules-25-03107-f003:**
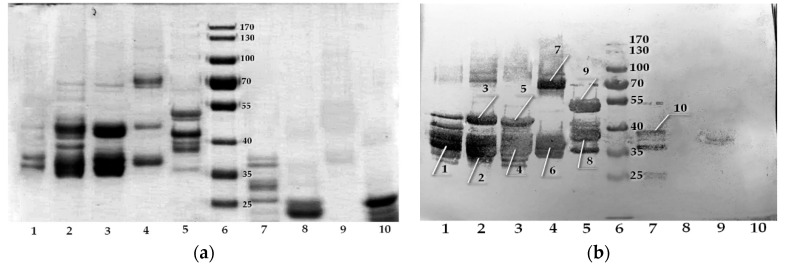
Electrophoresis of various prolamins (**a**) 1—*Triticum urartu*, 2—*Triticum turgidum*, 3—*Triticum aestivum*, 4—*Secale cereale*, 5—*Hordeum vulgare*, 6—molecular weight marker, kDa, 7—*Avena sativa*, 8—*Zea mays*, 9—*Dasypyrum villosum*, 10—*Setaria italica*; (**b**) western blotting of X6. Lines indicate the bands analyzed by MALDI-TOF, the results of which are shown in Table 4.

**Figure 4 molecules-25-03107-f004:**
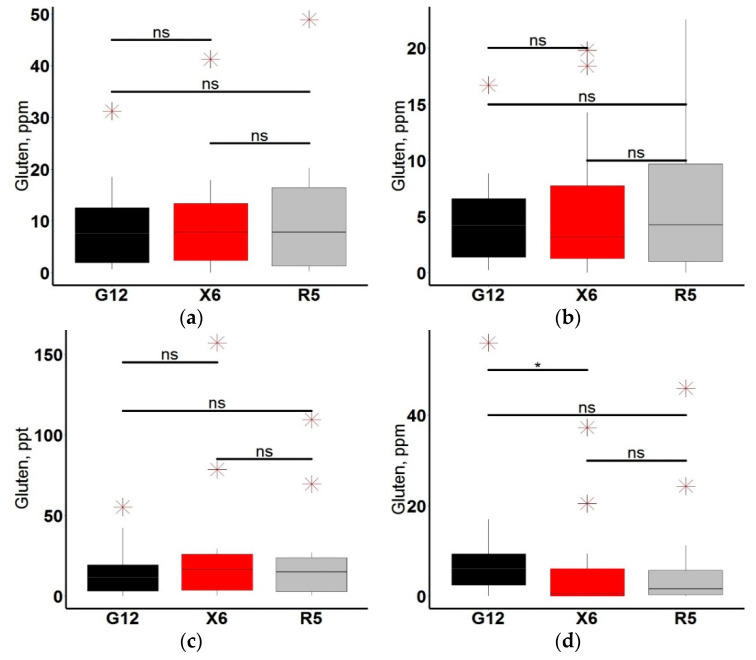
Gluten content in various groups of products: (**a**) bakery (*n* = 16), (**b**) flour (*n* = 16), (**c**) high–gluten products (*n* = 12), (**d**) snack (*n* = 18). Outliers are marked with a red asterisk, Mann–Whitney test significance levels are black: ns: *p* > 0.05; *: *p* ≤ 0.025.

**Figure 5 molecules-25-03107-f005:**
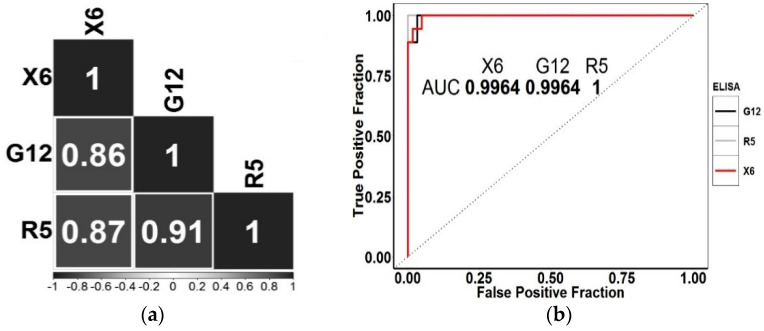
Gluten content comparison results obtained using various methods: (**a**) correlogram obtained by analyzing the gluten content using various test systems. The squares indicate the value of the Pearson correlation coefficient; (**b**) ROC curves obtained from the analysis of products using various ELISA.

**Table 1 molecules-25-03107-t001:** Commercial antibodies specificity to gluten.

Antibody	Protein’s Specificity	Epitope’s Specificity [7]
401.21	Main: HMW and LMW glutelin ω-gliadin Slight: α- and γ-gliadins [18]	Main: PQPQPFPQE PQQPPFPEE
R5	Main: α- and γ-gliadins Slight: ω-gliadin [18]	Main: QQPFP
G12	33-mer α2-gliadin [7]	Main: QPQLPY

**Table 2 molecules-25-03107-t002:** ELISA procedures.

Step	X6 Method	R5 Method	G12 Method
Extract dilution with ELISA buffer (extract + buffer, µL)	1:20 45 µL + 855 µL	1:12.5 80 µL + 920 µL	1:10 100 µL + 900 µL
Sample incubation	100 µL/well 30 min at 37 °C	100 µL/well 30 min at RT	100 µL/well 20 min at RT
Washing	Discard the solution from wells		
	3 times 300 µL/well	3 times 250 µL/well	5 times 300 µL/well
Incubation with conjugate	100 µL/well 30 min at RT	100 µL/well 30 min at RT	100 µL/well 20 min at RT
Washing	Discard the solution from wells		
	5 times 250 µL/well	3 times 250 µL/well	3 times 300 µL/well
Substrate	100 µL/well, Single mix	50 µL chromogen + 50 µL Substrate per well	100 µL/well, Single mix
Color development	15 min in the dark at RT	30 min in the dark at RT	20 min in the dark at RT
Stopping the reaction	100 µL/well Stop reagent		
Spectrophotometry	450 nm		

**Table 3 molecules-25-03107-t003:** OD values and S/N ratios obtained for various antibody pairs.

Tracer Antibody	Coating Antibody						Gliadin, ng/mL
	X4		X6		X10		
X4	0.756	0.756	0.128	0.124	0.341	0.332	0
	1.009	1.029	0.777	0.79	0.926	0.964	20
S/N ratio	1.35		6.22		2.81		
X6	0.238	0.257	0.098	0.101	0.156	0.167	0
	0.969	0.908	1.197	1.194	0.685	0.732	20
S/N ratio	3.79		12.02		4.39		
X10	0.545	0.598	0.646	0.656	0.558	0.488	0
	0.99	0.948	1.129	1.029	1	0.943	20
S/N ratio	1.7		1.66		1.86		

**Table 4 molecules-25-03107-t004:** X6 antibody specificity based on MALDI-TOF data.

No. band	Approx. kDa	Source	Protein	Protein Score	Sequence (NCBI Accession)	QXQPFPXP
1	37	*T. urartu*	γ-gliadin	121	ACJ03501.1	QPFPQP
2	36	*T. turgidum*	γ-gliadin	104 1	ACJ03444.1	QPFPQP
3	45	*T. turgidum*	LMW-glutenin	64 1	CAD61021.1	PQPFPQ
4	36	*T. aestivum*	γ-gliadin	120	QEH60939.1	QPFPQP QQPFP
5	45	*T. aestivum*	LMW-glutenin	133	AAV91998.1	QQPFPQ
6	35	*S. cereale*	γ-prolamin	110	AEW46841.1	QQPFPQ
7	70	*S. cereale*	75 K-secalin	105	ADP95485.1	PQQPFPQ QPFPQP
8	38	*H. vulgare*	γ-hordein	102	CAE45747.1	PQQPFP QPFPQP
9	50	*H. vulgare*	B-hordein	95	ACU09490.1	QPFPQ
10	39	*A. sativa*	avenin	154	CBL51496.1	QEQPFV QQQPFV

^1^ proteins identified using ‘ion score’ option.

**Table 5 molecules-25-03107-t005:** Determination of gliadin content in various cereals ethanolic extracts (ppm) using X6 antibody-based ELISA.

Protein	Source	Protein Concentration, mg/mL	ppm
Gliadin	*Triticum aestivum*	1	124,654.1
Avenin	*Avena sativa*	1	113.5
Hordein	*Hordeum vulgare*	1	141,809.8
Secalin	*Secale cereale*	1	105,209.2

**Table 6 molecules-25-03107-t006:** Gluten quantification in “snack” group (*n* = 18).

	*n* Total Positive	*n* Total Negative	McNemar’s Test *p*–Value (*p* < 0.05 Is Significant)	
G12	1	17	X6 vs. G12	1
X6	2	16	X6 vs. R5	NA
R5	2	16	G12 vs. R5	1

**Table 7 molecules-25-03107-t007:** McNemar’s test for all samples (*n* = 80).

X6				
R5		**Negative**	**Positive**	**McNemar’s *p*-value**
	Negative	62	0	0.48
	Positive	2	16	
X6				
G12		**Negative**	**Positive**	**McNemar’s *p*-value**
	Negative	64	1	1
	Positive	0	15	
R5				
G12		**Negative**	**Positive**	**McNemar’s *p*-value**
	Negative	62	0	0.25
	Positive	3	15	

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
