# Peer review of "X6: A Novel Antibody for Potential Use in Gluten Quantification"

_molecules, 2020, doi:10.3390/molecules25143107_

Round 1

Reviewer 1 Report

Shatalova et al. present a set of results that demonstrate that the X6 antibody can be used in an ELISA format in the context of gluten quantitation in gluten-free products. It presents results comparable to the Codex Alimentarius standard antibodies, R5 and G12. In general, the techniques used and the results achieved support the authors' claims. However, some sections are poorly described or discussed and the article would benefit from a more careful editing of the text and English. The authors declared important conflicts of interest, nevertheless the results seem clear and consistent. Supplementary results should also be better presented and captions provided. Some references are missing. Some points are explained in more detail:

Abstract - the background is missing.

Line 20-21 - Please improve the sentence and its content. Reference is missing.

Line 22 – Glutenins soluble in alkalis? Again, reference is missing for the whole paragraph.

Line 24 – oats? the most recent bibliography on the topic should be consulted. The reaction that oats can induce is thought to be the result of contamination of cereals such as wheat, barley or rye, rather than their own proteins.

Line 26 – Wrong. Please reformulate. All enzymes are proteins, but not all proteins are enzymes.

Line 28 – DQ2/DQ8 heterodimeric surface receptors.

Line 31 – pathogenesis of celiac disease.

Line 32 – saturated? Rich maybe

Line 35 – remove second α2-gliadin

Line 36 – italicize in vitro

Line 44 – mass spectrometry, MALDI-TOF? what's the difference?

Line 46 - reference

Line 47 – delete also

Line 50 – Skerrit et al. (date) or reference at the end of the sentence.

Line 56 – antibody

Line 58 – improve sentence

Line 62 – there is also

Line 63 – remove gliadin

Line 58 – 65 – Multiple references are missing.

Line 66 – has also

Line 66 – delete great

Line 67 – delete in products

Line 68 – “forming strong molecular scaffolds”. Not sufficient, please describe briefly the physical-chemical changes induced by food processing. Reference…

Line 80 - limit of quantification (LOQ)… the same for LOD; more care in the construction of sentences.

Line 78 – 82 – poorly written; please reformulate this paragraph and insert reference whenever necessary.

Line 85 – (…) (Zedira, G051), guanidine

Line 90 – XEMA (city, country)

Line 92 – “The apparently gluten-positive samples were not under the consideration.” Please explain better.

Supplementary S1 - Please provide a proper file, with a caption.

Line 101 – mAbs: Were antibodies developed by the authors or are they commercially available (XEMA)? In fact, the title and the abstract can lead to a wrong perception. Please make this clearer even if only in the material and methods.

Supplementary S3 – Please provide more information about these results. As previously mentioned, please provide also a proper file. S3 is referred first than S2…

Line 111 – Please provide more details on microarray

Line 120 – delete Preliminary or start another way

Line 121 – described anywhere… where? There is no reference. Elsewhere instead.

Line 136 – titration?

Line 151 -  “which as known  interferes with further staining” delete or put a reference on it!

Line 152 – in 40% met…, 10 % acetic…

Line 153 – 400 rpm; please refer the instrument.

Electrophoresis of prolamins – “Grinded cereals were extracted with 60% ethanol with the add of 0.7 M 2ME” = gliadins + glutenins, not only gliadins

Western-blotting of prolamins – “antibody-HRP conjugate” which antibody? Please provide more information.

MALDI-TOF – “Protein bands, which were reactive in Western-blot, were cut from the gel for MALDI-TOF analysis. The procedure was conducted at the Institute of Biomedical Chemistry, Moscow.” Insufficient, please provide more information.

Line 170 – “replaced with an in-house modified extraction method based on cocktail and ethanol combination”. Please provide more information.

Line 171 – food grinder. Please provide the instrument information.

Line 175 – clarified?

Line 176 – delete certain; procedure decribed…

Gluten measurement in foods with ELISA - Were the samples for R5 and G12 extracted differently than for X6? Please clarify. Why X6 was selected es “(X1–26, XH1 and XH4, mAbs)”? Please clarify. This section, material and methods, must be understandable for a wide audience and that allows a clear understanding and even the reproduction of all experiences.

Table 2 – how the 1:20 dilution was obtained? Please provide more information.

Statistical analysis – Please provide which statistical analyses were performed in R.

Line 184 – delete total

Line 189 – Based on

Line 190 – delete respectively or clarify

Line 193 – delete total

Line 197 – 199 - Improve this paragraph, poorly written.

Line 203 – at a specific position

Line 203 – led to a significant decrease

Line 207 – position 1

Figure 2 – Please provide full detail caption for figure a. figure b, in my opinion, y axis should be limited to 19. Also, the caption for figure b should be improved, is not clear.

Line 215 – Based on

Line 219 – Just four species are referred in materials and methods “T. aestivum, A. sativa, S. cereal, H. vulgare”. Please introduce all the species tested.

Figure 3 - Figure 3 can have a better presentation; numbers must be placed under the respective lanes. Molecular weights can also be placed in the figure.

Table 3 – Score for each identification is missing. More details about MALDI-TOF analysis too.

Table 4, line 233 to 236 - I understand the discussion, but in fact there is a lot of discussion about the role of avenins in celiac disease. The current opinion says that the immunogenicity of this species comes from some type of contamination with other cereals. The most recent literature should be referenced and this subject better discussed. In addition, does table 4 result from measurement with X6? This should be mentioned in the legend.

Line 242 – “X6-based ELISA is similar to ‘golden standards’”. Please consider to change gold standards by Codex Alimentarius standards for example.

X6-based ELISA is similar to ‘golden standards’ – There is new data about the use of antibodies to quantify gluten in gluten-based samples modified by different physical treatments. “results seem to clearly point out that the measurement of the potential immunogenicity of gluten using antibody assays should be performed with previously hydrolysed material.” because gluten solubility interferes with R5 method https://doi.org/10.1016/j.foodchem.2019.124986. Please comment about X6. Although the authors used different products and obtained similar results with different antibodies, quantification in hydrolyzed material was not carried out.

Line 243 – 247 – Reference??

Line 267 – delete a great distinction; a significant difference was observed

Line 268 – 270 - This conclusion sounds a bit forced. the values for a, b and d are also very low and very similar for the three antibodies tested.

Line 270 – but no difference was observed in relation to the R5 antibody.

Line 271 – 274 - What is the purpose of this test if nothing has been discussed?

Line 279 – 281 – “We excluded from the correlation analysis products with values obtained less than 5 ppm of gluten and containing more than 80 ppm (see Supplementary S2). The gluten content in the remaining samples (19 in total) was log-transformed for normalization.” Why have some values been transformed and others not? Doesn't this bias the analysis?

Line 284 – delete sentence; redundant

Line 289 – delete In conclusion

Line 308 – potential use in gliadin content

Line 312 – “we found the almost complete their coincidence.” Some sentences need to be carefully edited.

Line 314 – only the ELISA systems was tested

Author Response

Dear Sir / Madam,
Thank you very much for Your useful comments on our article.

-First of all, we tried to improve the English language of the article
-We also added references in all places where it was necessary
Lines 27-28: the paragraph in the Introduction was corrected, which, as it turned out when referring to the original text of the article, was the result of an incorrect translation

As far as we understood, the main questions concerned the design of the experiment, in particular, those aspects that related to ELISA, as well as the MALDI-TOF. As a result, we substantially revised the description of the methods and introduced the missing elements.

Line 88 M&M/ The detailed list of the antibody used for the experiments is incleded

lines 100-113 2.1 Antibody selection describes the analysis used for antibody selection via its reaction with 26/33-mer peptides 

line 124 2.2 Epitope Mapping/ Supplementary file was improved with the raw data provided by PEPperPRINT

lines 127-144 2.3 ELISA pre-construction now reflects the pipeline we used for choosing the antibody pair to work with.

lines 145-165 2.4 Determination of ELISA characteristics. We have included this paragraph to describe the methodology of the estimation of metrological characteristics for the ELISA constructed 

lines 177-178 The list of the grains used in immunoblotting study has been included

lines 206-220 2.7 MALDI-TOF/The description for MALDI-TOF procedure has been included

lines 224-227 2.8 Gluten measurements/ The important details of the assay applied are included into the section

lines 238-242 2.9 Statistical Analysis/ The description of the statistical methods alongside with the software used is included

lines 245-254 3.1 PFP motif is critical for antibody recognition/ The details of the antibody selection process for subsequent experiments is now described in details. We also include a list of antibody characteristics provided by the manufacturer as Supplementary File S3.

line 301 Table 4/ The results of the MALDI-TOF were supplemented with the information about protein scores.

line 308 Table 5/ Caption is clarified with the information about the ELISA used

line 322-325 3.3 X6-based antibody.../ The section is supplemented with the metrological characteristics of the ELISA constructed

line 350-351 The results of McNemar test are decribed

You also asked some questions in the review.

"Table 4, line 233 to 236 - I understand the discussion, but in fact there is a lot of discussion about the role of avenins in celiac disease. The current opinion says that the immunogenicity of this species comes from some type of contamination with other cereals. The most recent literature should be referenced and this subject better discussed. "

we consult the literature and found that the safety of the oats is still under the discussion. There are some species (very little) bearing epitopes, which could be recognized with G12 antibody doi:10.1038/srep42588. The reference is included in line 26. On the other hand, the discussion is mostly about the antibody affinity itself and influence of the certain amino acid sequence on it.

"There is new data about the use of antibodies to quantify gluten in gluten-based samples modified by different physical treatments. “results seem to clearly point out that the measurement of the potential immunogenicity of gluten using antibody assays should be performed with previously hydrolysed material.” because gluten solubility interferes with R5 method https://doi.org/10.1016/j.foodchem.2019.124986. Please comment about X6. Although the authors used different products and obtained similar results with different antibodies, quantification in hydrolyzed material was not carried out."

Now we are under development of the competitive ELISA using X6 antibody. Honestly, all of the antibodies selected during screening on peptides (X4/6/10) are under consideration. For this reason we now do not have enough data about the its comparison with standard ELISAs. Regarding to the article You provided We think that, of course, all type of treatments influence the performance of the analysis. But we think that such an issue arises mostly from the protein/sample variations resulted from the specific processing.

For our further experiment we will consider these data and expand the number of samples with those of different treatments. Thank You very much for this comment!

"Line 279 – 281 – “We excluded from the correlation analysis products with values obtained less than 5 ppm of gluten and containing more than 80 ppm (see Supplementary S2). The gluten content in the remaining samples (19 in total) was log-transformed for normalization.” Why have some values been transformed and others not? Doesn't this bias the analysis?"

From the entire sample number we have selected only those, which were in the range of 5 to 80 ppm for all ELISAs. This range was selected as all three kits were able to precisely detect gluten concentration in it. The total number of samples was 19. For normalization the data were log-transformed, and the selected samples were tested in correlation analysis. The remaining samples (80-19 = 61) were not considered for this analysis as they were not in the range of 5-80 ppm.

Line 314 – only the ELISA systems was tested.

Corrected. The first idea was that lateral sticks for the gluten detection use sandwich format, as sandwich ELISA does, so this type of the analysis was also included as potential. 

Reviewer 2 Report

The paper reports the characteristics of a novel anti-gliadin antibody X6 and its potential use for gluten detection in ELISA test kits. While the study itself may be interesting, there are several points that compromise the overall quality of the paper.

First of all, it is acknowledged that the authors are, most likely all, non-native English speakers. However, the quality of the English language is insufficient and does not help when trying to understand what was done here. This paper needs to be revised substantially based on languages issues alone.

The Introduction clearly lacks scientific background knowledge. L26 is wrong. L32, What do you mean with “saturated with proline and glutamine”?

A clearly formulated aim of the study is missing.

M+M/R+D: The authors do not explain how they arrived at the monoclonal antibodies X1-X26 in the first place. However, this information would be essential to be able to judge the quality of the X6 antibody.

L186: Related to the above, it is not clear how you already selected X4, X6 and X10.

L217ff: Usually extensive testing of potential crossreactivity to other allergens needs to be performed. This was not done here and, therefore, the specificity of X6 cannot be judged.

The need for this novel antibody is also not critically discussed. If it essentially gives the same information as the already established R5 and G12 mAbs, then why do we need the X6?

Overall, this manuscript lacks essential information to be able to judge the validity of the work and see whether it was carried out with appropriate rigor. There are very clear guidelines and recommendations what needs to be done in terms of validation before putting a new antibody out there. This was not done here, or at least it is not reported.

Author Response

Dear Sir / Madam,
Thank you very much for Your useful comments on our article.

-First of all, we tried to improve the English language of the article
-We also added references in all places where it was necessary
Lines 27-28: the paragraph in the Introduction was corrected, which, as it turned out when referring to the original text of the article, was the result of an incorrect translation

As far as we understood, the main questions concerned the design of the experiment, in particular, those aspects that related to ELISA, as well as the MALDI-TOF. As a result, we substantially revised the description of the methods and introduced the missing elements.

Line 88 M&M/ The detailed list of the antibody used for the experiments is incleded

lines 100-113 2.1 Antibody selection describes the analysis used for antibody selection via its reaction with 26/33-mer peptides 

line 124 2.2 Epitope Mapping/ Supplementary file was improved with the raw data provided by PEPperPRINT

lines 127-144 2.3 ELISA pre-construction now reflects the pipeline we used for choosing the antibody pair to work with.

lines 145-165 2.4 Determination of ELISA characteristics. We have included this paragraph to describe the methodology of the estimation of metrological characteristics for the ELISA constructed 

lines 177-178 The list of the grains used in immunoblotting study has been included

lines 206-220 2.7 MALDI-TOF/The description for MALDI-TOF procedure has been included

lines 224-227 2.8 Gluten measurements/ The important details of the assay applied are included into the section

lines 238-242 2.9 Statistical Analysis/ The description of the statistical methods alongside with the software used is included

lines 245-254 3.1 PFP motif is critical for antibody recognition/ The details of the antibody selection process for subsequent experiments is now described in details. We also include a list of antibody characteristics provided by the manufacturer as Supplementary File S3.

line 301 Table 4/ The results of the MALDI-TOF were supplemented with the information about protein scores.

line 308 Table 5/ Caption is clarified with the information about the ELISA used

line 322-325 3.3 X6-based antibody.../ The section is supplemented with the metrological characteristics of the ELISA constructed

line 350-351 The results of McNemar test are decribed

You also asked some questions in the review.

"L217ff: Usually extensive testing of potential crossreactivity to other allergens needs to be performed. This was not done here and, therefore, the specificity of X6 cannot be judged"

The specificity data is provided in Supplementary File S1 where the list of the products tested is described. The group 'Unprocessed' was not included into the Mann-Whitney test, because only three products showed gluten concentration above LOQ (5ppm), but it was included into the McNemar test, which was done for qualitative assesment of the ELISA.

"The need for this novel antibody is also not critically discussed. If it essentially gives the same information as the already established R5 and G12 mAbs, then why do we need the X6?"

Unfortunately, due to some commercial restrictions, the ELISA's based on G12 and R5 are not always available in, for example, our country. For this reason the development of the analogues, which are capable to quantify gluten in products, as preciesly as Codex's Standarts does, in our opinion, is actual. On the other hand, there are known problem with even perfect R5 antibody, giving positive results with soy proteins and lupins. We hope, that the X6, having larger recognizible epitope, can improve food control procedure for gluten content.

Round 2

Reviewer 1 Report

The authors provided important changes to the manuscript that made it ready to be accepted after minor changes. Some comments were ignored, but they are important to achieve the journal's publication standards.

Abstract - the background is missing. From Molecules website: “The abstract should be a single paragraph and should follow the style of structured abstracts, but without headings: 1) Background: Place the question addressed in a broad context and highlight the purpose of the study; 2) Methods: Describe briefly the main methods or treatments applied. Include any relevant preregistration numbers, and species and strains of any animals used. 3) Results: Summarize the article's main findings; and 4) Conclusion: Indicate the main conclusions or interpretations.”

Abstract – please substitute “golden standards”

Line 22 –soluble in dilute acid or alkali solution

Line 36 – rich in proline…

Line 47 – MALDI-TOF is a mass spectrometry technique; or mass spectrometry, such as MALDI...

Line 81 – to meet the requirements

Figure 2 – Please provide full detail caption for figure 2. Asterisks, ns…

Figure 3 - The appearance of figure 3 can be improved. Numbers below the lanes to identify the different samples and the molecular weight can be added to the figure.

Table 4 – Present as just Score. Threshold should be referred in the materials and methods. For instance, why different thresholds (55 and 82)?

Line 352 – ELISA; no plural in abbreviations.

Conclusion - Our data allow us to conclude that the sandwich ELISA method based on the X6 antibody has the potential to be used in test systems designed to determine the gluten content in foods.

Author Response

Dear Sir/Madam,

thank You for the comments regarding our article. Here the list of the corrections applied:

  • we have improved the abstract considering Your corrections;
  • we have corrected writing mistakes;
  • the caption for figure 2 is now supplemented with the comments;
  • we tried to improve the appiarence of the Figure 3: each protein line was numbered according to the sample. Western Blot: arrows were replaced with lines. We cannot delete it at all, bacause in our opinion the use of 'numbers only' format will lead to misreading of the picture;
  • MALDI-TOF: we consulted with the collegues, who conducted the analysis, about tresholds. As we were informed, different tresholds is a result of the certain options used for identification of individual protein. Most of the bands (8) were identified with Mascot's 'protein fingerprint' option, and they have the treshold of 82. For the identification of the remaining two samples they used 'ion score' option. The clarifiaction is included into the 2.7 MALDI-TOF section and into the footer of table 4.
  • Conclusion was also corrected.